# Induction of Human Wharton’s Jelly of Umbilical Cord Derived Mesenchymal Stem Cells to Be Chondrocytes and Transplantation in Guinea Pig Model with Spontaneous Osteoarthritis

**DOI:** 10.3390/ijms25115673

**Published:** 2024-05-23

**Authors:** Gulrez Nadeem, Kasem Theerakittayakorn, Sirilak Somredngan, Hong Thi Nguyen, Traimat Boonthai, Worawalan Samruan, Ponthep Tangkanjanavelukul, Rangsun Parnpai

**Affiliations:** 1Embryo Technology and Stem Cell Research Center, School of Biotechnology, Institute of Agricultural Technology, Suranaree University of Technology, Nakhon Ratchasima 30000, Thailand; gulrez.nadeem@yahoo.com (G.N.); kasemtheera@gmail.com (K.T.); sirilak.jss@gmail.com (S.S.); hongcn50c@gmail.com (H.T.N.); boonthait@gmail.com (T.B.); wsamruan@gmail.com (W.S.); 2School of Orthopedic Surgery, Institute of Medicine, Suranaree University of Technology, Nakhon Ratchasima 30000, Thailand

**Keywords:** umbilical cord, Wharton’s jelly, mesenchymal stem cells, osteoarthritis, guinea pig model

## Abstract

Osteoarthritis (OA) is a degenerative joint disease commonly found in elderly people and obese patients. Currently, OA treatments are determined based on their condition severity and a medical professional’s advice. The aim of this study was to differentiate human Wharton’s jelly-derived mesenchymal stem cells (hWJ-MSCs) into chondrocytes for transplantation in OA-suffering guinea pigs. hWJ-MSCs were isolated using the explant culture method, and then, their proliferation, phenotypes, and differentiation ability were evaluated. Subsequently, hWJ-MSCs-derived chondrocytes were induced and characterized based on immunofluorescent staining, qPCR, and immunoblotting techniques. Then, early-OA-suffering guinea pigs were injected with hyaluronic acid (HA) containing either MSCs or 14-day-old hWJ-MSCs-derived chondrocytes. Results showed that hWJ-MSCs-derived chondrocytes expressed specific markers of chondrocytes including Aggrecan, type II collagen, and type X collagen proteins and *β-catenin*, *Sox9*, *Runx2*, *Col2a1*, *Col10a1*, and *ACAN* gene expression markers. Administration of HA plus hWJ-MSCs-derived chondrocytes (HA-CHON) produced a better recovery rate of degenerative cartilages than HA plus MSCs or only HA. Histological assessments demonstrated no significant difference in Mankin’s scores of recovered cartilages between HA-CHON-treated guinea pigs and normal articular cartilage guinea pigs. Transplantation of hWJ-MSCs-derived chondrocytes was more effective than undifferentiated hWJ-MSCs or hyaluronic acid for OA treatment in guinea pigs. This study provides a promising treatment to be used in early OA patients to promote recovery and prevent disease progression to severe osteoarthritis.

## 1. Introduction

Osteoarthritis (OA) is prevalent among elderly individuals, obese patients, and physically active people who exert significant stress on their knee joints. The incidence of osteoarthritis increases with age and obesity. According to the World Health Organization’s (WHO) Global Burden of Disease Study 2010, over 70 million Europeans suffer from osteoarthritis, which hampers normal joint function due to the low self-repairing capability of cartilage. Knee and hip osteoarthritis are also causes of disability [1]. Osteoarthritis can be triggered by a range of factors, including non-genetic factors, e.g., age, sex, occupational activities, sports activity, high body mass index, obesity, diabetes mellitus, muscle weakness, mechanical instability, bone marrow lesion, and bone mineral density [2,3]. Genetic factors may also play a role, such as changes in gene expression in cartilage and subchondral bone. These factors can impact joints in the body, particularly in the hands and limbs that are subjected to greater weight-bearing stress, resulting in pain and functional impairment in adults. Initial symptoms involve deterioration of articular cartilage leading to pain, bone dysfunction, and difficulties in performing daily activities. Articular cartilage is a specialized type of connective tissue composed of cartilage cells and typically found in synovial joints. These cells produce extracellular matrix (ECM) and preserve the function of the tissue. Articular cartilage does not possess self-healing abilities due to the absence of blood vessels, lymphatic vessels, and the nervous system [4]. Arthritic cartilage degeneration can cause various symptoms, including growth abnormalities in children, injuries caused by stress from trauma, and age-related osteoarthritis [5].

Current OA treatments are commonly determined based on disease severity and the physician’s recommendations, including pharmacological and non-pharmacological therapies. The first-line pharmacologic treatment is non-steroidal anti-inflammatory drugs (NSAIDs) to cure patients with hand, knee, and/or hip OA, particularly oral NSAIDs being the initial medical choice in the OA treatment and being recommended over other available oral medications [6]. However, NSAIDs’ prescription should be considered with caution due to their gastric ulcer complication and cardiovascular risk [7]. A combination of pharmacological and non-pharmacologic treatment, namely, diet and weight loss, physical therapy and exercise, and nutritional supplements (glucosamine and chondroitin sulfate) is common advice for OA treatment to reduce symptoms and improve the functional performance of the joint. Surgery is an invasive procedure that should only be conducted when the combined therapy is unsuccessful in producing the desired outcomes [7]. Additionally, there is a possibility of recurrence and complications after surgery in many patients. Consequently, a novel and more effective procedure for osteoarthritis is indispensable, like the application of cartilage cells. However, the extraction of these cells from a human requires invasive surgery, which is complicated and expensive [8,9]. Research is currently underway to explore the potential use of mesenchymal stem cells (MSCs) in treating osteoarthritis. The stem cells possess the unique ability to stimulate the growth of cartilage cells and other types of cells. MSCs are utilized in treating various disorders and can be sourced from several different locations, including bone marrow, blood, adipose tissue, and dental pulp. They can be isolated and cultured with a high level of proliferation activity. Previous studies have identified Wharton’s jelly, found in the umbilical cord of humans, as a common source of MSCs. This tissue can be collected from pregnant women following childbirth, without requiring a complex collection process [10]. As a result, MSCs isolated from the Wharton’s jelly of human umbilical cords are a promising area of interest for future clinical trials.

The use of Dunkin Hartley guinea pigs as an animal model for studying spontaneous cartilage degeneration in the knee joint, which is similar to osteoarthritis in humans, has been well established [11,12,13,14]. Researchers have reported that the knee joint of guinea pigs closely resembles that of humans affected by osteoarthritis [11,12]. Moreover, spontaneous cartilage degeneration in Dunkin Hartley guinea pigs was used for study [15]. Previous studies have demonstrated that injecting mesenchymal stem cells (MSCs) with hyaluronic acid (HA) into the articular cartilage of guinea pigs with osteoarthritis led to recovery [16]. HA-based formulations are currently delivered into the joint to relieve pain and improve the joint mobility of OA patients by partial restoration of the rheological properties of the synovial fluid [17].

In this study, we isolated MSCs from human Wharton’s jelly of the umbilical cord and induced them into cartilage cells. We then transplanted the early chondrogenic differentiated MSCs into guinea pigs which have osteoarthritis and monitored their progress to evaluate the effectiveness of the treatment. The results demonstrated promising outcomes in the experimental animals, suggesting that this treatment approach using early chondrogenic differentiated MSCs could be developed into a viable treatment option for patients with osteoarthritis. Furthermore, this method is simpler and less invasive than surgical treatments, making it a potentially safer option for patients. Therefore, the purposes of this study were to differentiate human Wharton’s jelly-derived mesenchymal stem cells (hWJ-MSCs) isolated from human umbilical cord tissues into chondrocytes and characterize hWJ-MSCs-derived chondrocytes prior to transplantation in OA-suffering guinea pigs.

## 2. Results

### 2.1. Isolation and Characterization of MSCs

hWJ-MSCs were obtained from two freshly collected umbilical cords (named as WJ01 and WJ07) at Maharat Nakhon Ratchasima Hospital, Thailand. The characteristics of hWJ-MSCs, including cell surface protein expression, colony-forming units, population doubling time, and differentiation ability, were determined.

Colony-forming unit results from WJ01 and WJ07 cell lines at passage 4, 5, 6, 7, and 10 are shown in Figure 1A. Colony-forming unit results of WJ01 at passage 4, 5, 6, 7, and 10 were between 18.17 ± 2.08 and 24.67 ± 5.48. Colony-forming unit results of WJ07 at passage 4, 5, 6, 7, and 10 were between 35.17 ± 2.08 and 38.17 ± 1.76. Thus, the results of the colony-forming unit of the WJ01 cell line were significantly lower than WJ07 cell line in all passages.

Population doubling time (PDT) was comparable between the two cell lines (Figure 1B). The PDT of both cell lines at P4–P7 and P10 varied from 38.47 ± 4.16 h to 51.30 ± 1.72 h. There was no significant difference between the two cell lines. Both hWJ-MSCs were positive for CD73, CD90, and CD105 and negative for CD34 and CD45 (Figure 1C). However, the proportion of WJ07 cells positive for CD105 was very low (32.01%). Both cell lines had adipogenic and osteogenic induction abilities (Figure 1D). Lipid droplets were much more detected in WJ07 cells than in WJ01, but the osteogenic differentiation potentials of both cell lines were similar.

There is no obvious difference between the adipogenic and osteogenic differentiation ability of the two cell lines, but only the WJ01 cell line qualified the standard of MSCs’ surface markers. WJ07 had higher CFU-F but a lower expression of CD105 than WJ07, while no difference in PDT was found between the two groups. Therefore, the WJ01 cell line was used for transplantation into guinea pigs with osteoarthritis.

### 2.2. Characterization of Chondrocytes Derived from hWJ-MSCs

After 28 days of chondrogenic induction, Sox9 (early chondrocyte stage) and type II collagen (mature chondrocyte stage) were highly expressed in chondrocytes derived from MSCs cells (Figure 2A,C). However, Aggrecan (mature chondrocyte stage) and type X collagen (hypertrophic chondrocyte stage) were low in expression in these differentiated cells (Figure 2B,D).

Gene expression was examined by qPCR at day 0, 3, 7, 14, and 28 of chondrogenic differentiation and compared with cartilage cells isolated from dissected human knee cartilages (positive control; Figure 3). *β-catenin* gene expression of chondrogenic differentiated cells at day 3, 7, and 14 was not significantly (*p* > 0.01) different but was significantly (*p* < 0.01) higher than undifferentiated MSCs (day 0). Levels of *β-catenin* gene significantly (*p* < 0.01) increased in the cells at day 28, compared to the cells at day 0, 3, 7, and 14 and the positive control (chondrocyte). Moreover, the *β-catenin* gene expression of the cells at day 3 and 14 was also significantly (*p* < 0.01) higher than the positive control. Expression of *Sox9* gene was significantly higher in induced chondrogenic cells at day 28 than those at day 0-14 (*p* < 0.01). At day 3, 7, and 14 of induction, *Runx2*, *Col2a1*, *Col10a1*, and *ACAN* gene expression showed no significant difference among groups. Until day 28, the expression of *Runx2*, *Col2a1*, *Col10a1*, and *ACAN* genes was significantly higher than the undifferentiated MSCs (day 0) but still significantly lower than the positive control.

Type II collagen proteins of chondrogenic differentiated cells at day 28 were compared to MSCs using an immunoblot technique. Pre-induced MSCs expressed very low levels of type II collagen, but differentiated chondrogenic cells expressed a very high level of type II collagen. At the same time, low levels of type X collagen expression were also found (Figure 4).

### 2.3. Chondrocyte Transplantation Results

Dunkin Hartley guinea pigs were used as animal models. Guinea pigs were divided into five groups: (1) 3-month-old guinea pigs with normal knee joints (normal), (2) 7-month-old guinea pigs with no treatment (spontaneous osteoarthritis; OA), (3) 7-month-old guinea pigs with HA injections, (4) 7-month-old guinea pigs with HA+MSCs injections, and (5) 7-month-old guinea pigs with HA+ differentiated chondrogenic MSCs. The differentiated chondrogenic MSCs were stained with CFDA-SE fluorescent dye before transplantation. After staining with CFDA-SE, the stained cells survived and grew normally when cultured (Figure 5). Intra-articular injection was performed in the knee joints of the guinea pigs. After transplantation, all the guinea pigs were healthy, and there were no observable abnormalities in the knee joints after the injection.

### 2.4. Macroscopic Examination Results

Five samples of the proximal tibia were obtained from each guinea pig group and dyed with India ink as shown in Figure 6. The rough cartilage surface clearly showed the black color of India ink particles. In group 1, there were no blacked out areas on the cartilage surfaces. Only the medial sides of the tibial cartilages in groups 2–5 of 7-month-old guinea pigs were blacked out. In the cartilage of group 2 (not transplanted), the blacked areas were wider than groups 3, 4, and 5. Degenerative symptom scores of each group are displayed in Figure 6F. The degenerative scores of the 7-month-old guinea pigs in groups 2–5 were significantly higher than group 1 (the 3-month-old guinea pigs; *p* < 0.01).

### 2.5. Histology Results

CFDA-SE fluorescence-stained cells in the knee sample of guinea pigs were examined using cryosection method and observed under a fluorescent microscope. Human nuclei were also stained with a red fluorescent antibody. Green fluorescent cells of CFDA-SE with red staining were observed in the human nuclei. This confirmed that the cells attached to the cartilage surface were human cells that were transplanted into the guinea pig’s knee joint (Figure 7). The results also pointed out that hWJ-MSCs-derived chondrocytes could adhere to the cartilage surfaces together with recovery of the damaged cartilage.

The knee joints of guinea pig were cryosectioned and examined histologically using H&E and Safranin O staining techniques (Figure 8 and Figure 9), respectively. A smooth cartilage surface, a uniform cartilage surface layer, and the extracellular matrix were present in group 1. However, the cartilage of 7-month-old guinea pigs in groups 2–5 showed cartilage imperfection at the medial side of the tibia indicated by unsmooth thickness of the cartilage surface layer, the rupture of the cartilage surface (the arrows in Figure 8 and Figure 9), extracellular matrix loss, and the formation of a cavity within the cartilage tissue. Group 2 had the most severe cartilage damage among all groups. Cartilage damage was reduced in group 3 (HA injection) and group 4 (HA injection with MSCs). On average, MSCs caused the least cartilage damage across all samples. In group 5 (HA injection with chondrogenic differentiated cells), the lowest cartilage damage was noticeable, compared to other groups. Cartilage damage scores based on the Mankin criterion procedure are illustrated in Figure 10. The knee joints of guinea pigs in groups 1, 2, 3, 4, and 5 showed cartilage damage scores as 1.6 ± 0.5, 6.6 ± 1.8, 6.4 ± 2.1, 4.8 ± 1.5, and 3.0 ± 1.9, respectively. The values in groups 2, 3, and 4 were significantly higher than group 1 (*p* < 0.01), but there was no difference between groups 5 and 1.

### 2.6. Immunohistochemistry Results

Immunohistochemistry was examined by a specific antibody against type II collagen (Figure 11). The area with a high type II collagen expression will be darkened. The cartilage of the group 1 showed high type II collagen expression, with a dark colored covering over the cartilage. Even though the cartilage of group 2–5 showed unequal expression, there was no difference between groups.

### 2.7. Immunoblot Results

Total proteins from guinea pig tibia cartilages were extracted and analyzed for type II collagen, type I collagen, MMP13, and β-actin proteins. The results were compared with human knee cartilage. Protein bands isolated by gel electrophoresis and immunoblot are shown in Figure 12. Intensity changes in type II collagen, type I collagen, and MMP13, versus β-actin protein acting as an internal control, are presented in Figure 13.

## 3. Discussion

hWJ-MSCs were obtained and expanded from two cell lines. The characteristics of both cell lines were determined using various methods such as CFU assay, PDT, MSC surface markers, and their capability to differentiate into osteocytes, adipocytes, and chondrocytes. In particular, the WJ01cell line had more than 95% of MSC surface markers, including CD73, CD90, and CD105, while negative markers (CD34 and CD45) were less than 2%. Moreover, the WJ01cell line demonstrated the ability to differentiate into osteocytes, adipocytes, and chondrocytes, in accordance with the standards set by the International Society for Cell and Gene Therapy [18,19,20]. Doubling times for WJ01 cell line were around 40–50 hrs for passages 4, 5, 6, 7, and 10, which were similar to MSCs obtained from the Wharton’s jelly tissue of human umbilical cords [21,22]. A specific MSC cell line was selected and utilized to generate cartilage cells over a period of 28 days. As per a previously established method [23], MSCs can be stimulated to differentiate into cartilage cells. The outcomes demonstrated the production of high levels of proteoglycan and strong Alcian blue staining in the generated cartilage cells. Additionally, immunofluorescence staining was performed to examine the expression of various proteins, such as Sox9, Aggrecan, type II collagen, and type X collagen [24,25]. The mature cartilage cells exhibited significant expression of the Sox9 and type II collagen proteins, while moderate expression of the aggrecan protein was observed. However, the findings indicated a low expression of type X collagen in the aged cartilage cells, with only minimal staining observed in the induced cells. Consistent with this observation, the results of protein quantification via immunoblot indicated a high expression of type II collagen and low expression of type X collagen. The expression of several genes was analyzed at different time points during the study. The *β-Catenin* gene expression demonstrated a rapid increase at day 3, indicating the involvement of the Wnt/β-Catenin signaling pathway in chondrogenic differentiation [26,27]. Similarly, the *Sox9* gene expression also displayed a quick increase, with the highest levels observed at day 28. However, the *ACAN*, *Col10a1*, and *Runx2* genes only exhibited a slight increase in expression at day 28. The current study compared the effects of cell transplantation for knee osteoarthritis between MSCs and MSC-derived chondrocytes. It was observed that the induction of cartilage cells led to the production of hypertrophic chondrocytes, a common type of cartilage cell found in osteoarthritis patients in the long term [28,29]. In this work, MSCs were induced into cartilage cells in the early stage of chondrogenic differentiation for 14 days. At day 14, the gene expression of *Sox9* and *β-Catenin* increased, while *Col10a1* and *Runx2*, both of which are associated with hypertrophic chondrocyte aging in cartilage, were not expressed [28,29]. In this study, Dunkin Hartley guinea pigs were used, as they are prone to developing osteoarthritis with age. The normal articular cartilage of three-month-old guinea pigs was compared to that of seven-month-old guinea pigs with early-stage osteoarthritis [12]. The knees of the guinea pigs were injected with either cell-free HA (Hyruan^®^III) or with either MSCs or chondrogenic differentiated cells. In this study, it was observed that seven-month-old guinea pigs that received only one injection of cell-free HA did not show any significant difference compared to the not-injected group. HA-based formulations, viscosupplements, are intended to recover the rheological properties of the synovial fluid, resulting in the improvement of pain and joint mobility [30]. Chemically modified HA improved mechanical performance during high-frequency solicitation and showed a prolonged viscosupplementation effect, compared to the unmodified, linear HA-based product [17]. However, the guinea pigs that received injections of HA containing either MSCs or early cartilage-differentiated cells showed a reduction in osteoarthritis. The injected cells adhered to the cartilage surface, thereby repairing the damaged cartilage and making it smoother, similar to normal cartilage. Histological analysis showed that the cartilage tissue in the injected cells had a smoother surface compared to the non-injected group. It was concluded that using HA containing early chondrogenic differentiated cells from MSCs could be an effective method for restoring articular cartilage and could be more effective than using MSCs alone for treating osteoarthritis.

Autologous chondrocyte implantation has proved to induce cartilaginous tissues in joint defects [31,32]. However, this technique has some limitations, such as technical complexity, costs of the two surgical procedures, de-differentiation of chondrocytes during in vitro expansion, and limited amount of cartilage from a small biopsy [33]. To solve these problems, allogeneic chondrocytes derived from cadaveric articular cartilage were implanted; this is preferred because it needs one surgery and reduces patient morbidity, and there is availability of a large chondrocyte depot [33]. Inflammatory responses made of xenogeneic/allogenic materials in the case of cell or organ transplantation are notable problems. Allogeneic chondrocytes seeded on xenogeneic scaffolds do not suppress graft inflammation but induce variable inflammatory responses involving mast cells and macrophages [34]. Although allogenic MSCs are shown to support cartilage regeneration and decrease the symptoms of OA [35], the inflammatory responses from xenogeneic MSCs and MSCs-derived chondrocytes remain poorly studied and should be further investigated.

## 4. Materials and Methods

### 4.1. Reagents

This study employed chemicals sourced from Sigma-Aldrich Corporation in St. Louis, MO, USA. Antibodies were obtained from Thermo Fisher Scientific located in Cleveland, OH, USA. Cell culture media were purchased from Gibco based in Paisley, UK, and plastic cell culture devices were procured from SPL Life Sciences in Gyeonggi-do, South Korea.

### 4.2. hWJ-MSCs Isolation and Culture

The human umbilical cords (n = 2) were obtained from Maharat Nakhon Ratchasima Hospital (Nakhon Ratchasima, Thailand) with the mother’s informed consent. The cords, which were approximately 7–10 cm in length, were washed using phosphate-buffered saline (−) ((PBS(−)). The hWJ-MSCs were isolated from the umbilical cord and cultured as previously described by Tanthaisong et al. (2017) [23]. Briefly, the gelatinous Wharton’s Jelly tissues were collected and sliced into small pieces (2–5 mm^2^). These pieces were placed in 90 × 15 mm culture dishes and grown in alpha modification of Eagle’s medium (α-MEM) enriched with 2 mM L-glutamine, 100 U/mL penicillin, 100 µg/mL streptomycin, and 10% fetal bovine serum (FBS). The MSCs were expanded until passage 3, cryopreserved with 10% dimethyl sulfoxide (DMSO) in culture media, and then stored in liquid nitrogen.

### 4.3. hWJ-MSCs Characterization

#### 4.3.1. Colony-Forming Unit (CFU) Assay

To evaluate the colony forming ability, the colony-forming unit assay was conducted by seeding 200 MSCs in a 6-well dish and culturing them for two weeks, with the medium being replaced every two days. The MSCs were fixed using 4% paraformaldehyde (PFA) for 20 min and then subjected to staining with 0.5% crystal violet to permit a visual assessment of the colony. The assessment was carried out for passages 4, 5, 6, 7, and 10 of MSCs. Subsequently, the stained cells were scrutinized using an inverted microscope (Eclipse Ti-S, Nikon Imaging Japan Inc., Tokyo, Japan) through the NIS-Elements D program (Nikon Imaging Japan Inc., Tokyo, Japan). An aggregate of no less than 50 cells was regarded as a colony. The numbers of colonies were then calculated following the equation below with each condition being tested thrice.
% CFU = (Total number of colony × 100)/Initial cells seeded (%)(1)

#### 4.3.2. Population Doubling Time (PDT)

Triplicates of cells ranging from passages 4 to 10 were seeded onto a 35 mm culture dish at a density of 4000 cells/cm^2^ and, then, cultured in α-MEM supplemented with 10% FBS. Following 72 h of culture, the number of viable cells was determined using 0.4% Trypan Blue staining. PDT was calculated using the formula below.
PDT = (t × log2/(logNF − logNI)(2)
where NI = Initial cells seeded, NF = Final numbers of cells, t = Time (hours).

#### 4.3.3. Flow Cytometric Analysis 

To verify the surface markers of MSCs, a flow cytometric analysis was carried out. In this analysis, MSCs at passage 5 were mixed with PBS(−) and incubated with various antibodies, including anti-CD73-APC, anti-CD90-APC/A750, anti-CD105-PE (dilution 1:100, Biolegend, San Diego, CA, USA), anti-CD34-PE (dilution 1:10, Beckman Coulter, Brea, CA, USA), and anti-CD45-FITC (dilution 1:20, Biolegend). As negative controls, isotype control antibodies were used. The incubation was carried out in the dark for 20 min, after which the samples were washed with PBS(−) and analyzed using an Attune^TM^ NxT Flow Cytometer (Attune^TM^ NXT, Thermo Fisher Scientific, Cleveland, OH, USA).

#### 4.3.4. Differentiation Ability

To induce osteogenic differentiation in MSCs, cells at passage 5 were cultured in 4-well culture plates coated with 0.1% gelatin until they reached 70% confluence. The induction medium contained α-MEM medium supplemented with 100 nM dexamethasone, 0.2 mM L-ascorbate-2-phosphate, 10 mM β-glycerophosphate, 100 U/mL penicillin, and 100 µg/mL streptomycin. The induction medium was replaced every 3 days, and the cells were cultured for 21 days. Calcium deposits from the cells were stained with Alizarin red and visualized under an inverted microscope.

To induce adipogenic differentiation in MSCs at passage 5, they were cultured in 4-well culture plates coated with 0.1% gelatin until they reached 70% confluence. The induction medium was α-MEM medium supplemented with 10 µM insulin, 100 µM indomethacin, 1 µM dexamethasone, and 0.5 mM isobutyl methylxanthine (IBMX). After 7 days of induction, IBMX was removed from the medium. The medium was replaced every 3 days, and the cells were cultured for a total of 21 days. Then, the cells were stained with Oil Red O to observe oil droplets, which were visualized under an inverted microscope.

#### 4.3.5. Chondrocyte Differentiation

MSCs at passage 5 were cultured in 4-well culture plates coated with 0.1% gelatin until they reached 70% confluence. The cells were then treated with an induction medium consisting of α-MEM medium supplemented with 10 µg/mL ITS-X, 50 µg/mL L-ascorbate-2-phosphate, 40 µg/mL L-proline, 100 µg/mL sodium pyruvate, 100 nM dexamethasone, 10 ng/mL TGF-β3, and 2% FBS. The induction medium was replaced every 3 days, and the cells were cultured for 28 days. The glycosaminoglycan extracellular matrix was evaluated using Alcian blue 8× staining and examined under an inverted microscope.

#### 4.3.6. Chondrocyte Characterization by Immunocytochemistry Staining (ICC)

Following 28 days of chondrocyte induction from MSCs, the cells were fixed in 4% PFA for 20 min, permeabilized, and blocked with a solution consisting of 5% BSA, 5% normal goat serum, and 0.1% Triton-X-100 at 37 °C for 1 h. Subsequently, the cells were incubated with specific primary antibodies, including anti-collagen type II anti-body (dilution 1:100), anti-collagen type X antibody (dilution 1:100), anti-Sox9 anti-body (dilution 1:100, obtained from Abcam, Cambridge, UK), and anti-Aggrecan anti-body (dilution 1:100, obtained from Abcam), at 4 °C overnight. On the following day, the cells were incubated with secondary antibodies, including Alexa fluor^®^ 594 goat anti-rat IgG (dilution 1:250, obtained from Invitrogen, Carlsbad, CA, USA) and Alexa fluor^®^ 488 goat anti-mouse IgG (dilution 1:1000, obtained from Invitrogen), for 2 h at room temperature. Finally, the cells were stained with 6-diamino-2-phenylindole (DAPI) at a dilution of 1:1000 and mounted with Vectashield antifade mounting medium (Vector Laboratories, Burlingame, CA, USA). Samples were observed using a fluorescence inverted microscope (Eclipse Ti-S, Nikon Imaging Japan Inc.) by the NIS-Elements D program (Nikon Imaging Japan Inc., Tokyo, Japan).

#### 4.3.7. Chondrocyte Characterization by Gene Expression Analysis

After MSCs were induced to undergo chondrogenic differentiation for 28 days, total RNA was prepared by a kit from RBC Real Genomics, RBC Bioscience based in Taipei, Taiwan. Then, cDNA synthesis was performed using oligo-dT primers and the iScript™ Reverse Transcription Supermix for RT-qPCR from BioRad, Hercules, CA, USA. To examine gene expression, KAPA SYBR-Green PCR Master Mix from Applied Biosystems, Carlsbad, CA, USA was used with the QuantStudio 5 real-time PCR system from Applied Biosystems, Carlsbad, CA, USA. The specificity of the specific primers (listed in Table 1) was confirmed by conducting a melting curve analysis [23]. The reference gene used to standardize the target genes was *GAPDH*, and the expression fold change was determined in relation to the control cells. To ensure accuracy, qPCR was performed three times, and relative changes in gene expression analysis were carried out using the 2^−∆∆CT^ method.

Chondrocytes obtained from human cartilage were also isolated and utilized as a positive control. It is worth noting that the use of chondrocytes derived from human cartilage was authorized by the Ethics Committee for Research Involving Human Subjects at Suranaree University of Technology, with the reference number EC-61-56.

#### 4.3.8. Chondrocyte Characterization by Western Blot Analysis

After 28 days of chondrogenic differentiation, the total proteins were extracted using a lysis buffer consisting of 10% sodium dodecyl sulfate (SDS), 0.1 M dithiothreitol (DTT), 1% glycerol, 1.2% urea, 1M Tris- HCl pH 7.4, along with a complete protease inhibitor. The total protein concentration was determined using the Bradford assay. Subsequently, 20 µg of total protein was separated using SDS-PAGE (10% resolving gel), and the separated protein was transferred onto PVDF membranes (Immun-Blot^®^ PVDF Membrane, Bio-Rad Laboratories). Next, the membranes were blocked in TBST (Tris-buffered saline with 0.1% Tween 20) containing 5% skim milk at room temperature for 1 h. For detection of collagen type II and type X proteins, membranes were incubated over-night at 4 °C with primary antibody solutions (1% BSA in TBS, Tris-buffered saline). The membranes were then washed with TBST and incubated at room temperature for 1 h with secondary antibodies conjugated with horseradish peroxidase (HRP; Abcam) that were diluted 1:2000 in 5% skim milk in TBST. The chemiluminescent substrate was added using an ECL substrate kit (Ultra-high sensitivity, Abcam) following the manufacturer’s instructions. Protein bands were visualized using ImageQuant^TM^ LAS 500 (GE Healthcare Life Sciences, Marlborough, MA, USA). For collagen type II and type X protein normalization, β-actin was used as a housekeeping control. Data were compared to negative control cells, and bands with saturated pixels were excluded.

### 4.4. Chondrocyte Transplantations

#### 4.4.1. Experimental Animals

Guinea pigs of 3 months’ age were identified as normal articular cartilage guinea pigs, while guinea pigs of 7 months’ age were identified as having minor OA. The study randomly assigned guinea pigs to five groups (N = 10/group) using a Completely Randomized Design (CRD) depicted in Figure 14: (1) normal group consisting of 3-month-old guinea pigs with no cells transplant, (2) OA group consisting of OA guinea pigs of 7 months with no cells transplant, (3) HA group consisting of OA 7-month-old guinea pigs who received HA injection, (4) HA+MSCs group consisting of OA 7-month-old guinea pigs who received HA injection with MSCs, and (5) HA+dif.MSCs group consisting of OA 7-month-old guinea pigs who received HA injection with differentiated chondrogenic cells. The guinea pigs were marked by notching on the pinna of the ear for identification among treatments. Carboxyfluorescein diacetate succinimidyl ester (CFDA-SE; Thermo Fischer Scientific, Waltham, MA, USA) was used to label cells before transplantation [21].

#### 4.4.2. Preparation of Chondrocytes Derived from MSCs

On day 14 of chondrogenic induction, chondrocytes derived from MSCs at passage 5 were collected for transplantation. The MSCs-derived chondrocytes were detached and separated by a 30 min digestion process using 0.2% collagenase type II and followed by 0.25% Trypsin. Before injection, these cells were labeled with CFDA-SE.

#### 4.4.3. Cell Transplantation

In group 4 and 5, each guinea pigs received a 100 µL injection containing a total of 1 × 10^6^ cells (MSCs in group 4, and chondrocytes derived from MSCs in group 5) that were labeled with CFDA-SE and suspended in HA (Hyruan^®^III; LG Chem, Seoul, Republic of Korea). The injection was administered into the medial compartment of the knees of the guinea pigs [16].

#### 4.4.4. Macroscopic Examination

At five weeks post-transplantation, the guinea pigs were euthanized using carbon dioxide fumigation. The proximal tibiae of each guinea pig were then opened, and their cartilage repair in the degenerative knee was examined [16]. This study used five proximal tibia samples to perform the macroscopic examination. The tibiae’s distal heads were first stained with India ink for a minute. The cartilage surfaces were then washed with PBS(−) before being examined and scored following the criteria in Table 2.

#### 4.4.5. Histology and Immunohistochemistry

Five proximal tibia samples were fixed in 10% buffered formaldehyde at 4 °C for 72 h. To decalcify the tissues, they were soaked in 5% nitric acid for 7 days and, then, embedded in optimal cutting temperature (OCT) compound. Tissue slices with a thickness of 15 mm were sectioned [39]. The tissue slides were stained using hematoxylin and eosin (H&E) and safranin O before being examined and scored following the Mankin et al. [40] and Armstrong et al. [41] methods. In the case of immunohistochemistry, the tissue slides were incubated using primary polyclonal antibodies (rabbits against type II collagen) with a 1:200 dilution (COSMO BIO, Tokyo, Japan) for an hour at room temperature. After washing with PBS, the tissue slides were incubated in HRP anti-mouse IgG secondary antibody for 30 min at room temperature. The tissue slides were visualized using VECTA STAIN ABC Reagent (Vectastain Elite Kit; Vector Laboratories). To conduct fluoroscopic analysis, CFDA-SE labeled cells were studied in a tibial frontal section using a fluorescence microscope with 492 nm and 517 nm settings [16].

#### 4.4.6. Immunoblot Analysis

Proteins were extracted from tissues and separated via SDS-PAGE using a 15% resolving gel. After electrotransfer to PVDF membranes (Immun-Blot^®^ PVDF Membrane), the membranes were blocked using a solution of 5% skim milk in TBST (Tris-buffered saline with 0.1% Tween 20) for 1 h at room temperature. To detect type I collagen, type II collagen, and matrix metalloproteinase-13 (MMP13 dilution 1:20), the membranes were incubated with primary antibody solutions (1% BSA in TBS Tris-buffered saline) overnight at 4 °C. After being washed with TBST, secondary antibodies (goat anti-rabbit or goat anti-mouse) conjugated with horseradish peroxidase (HRP; Abcam, Cambridge, UK) were applied for 1 h at room temperature at a dilution of 1:2000 in 5% skim milk in TBST. The chemiluminescent substrate was applied using an ECL substrate kit (Ultra-high sensitivity, Abcam, Cambridge, UK) following the manufacturer’s suggestions. The protein bands were then imaged using Image Quant^™^ LAS 500 (GE Healthcare Life Sciences, Marlborough, MA, USA), with β-actin used as a housekeeping control for normalization with type I collagen, type II collagen, and MMP13 protein [16]. This study compared data between the experimental animals and human cartilage. Additionally, the data from bands with saturated pixels were not quantified.

### 4.5. Statistical Analysis

Statistical analysis was conducted on three to five samples, and the data are presented as mean ± standard deviation (S.D.). To compare differences between the control and treated groups, a one-way analysis of variance (ANOVA) was employed, followed by the Tukey–Kramer Honest Significant Difference (HSD) as a post hoc test. Results with a *p*-value less than 0.05 were regarded as significant, whereas those with a *p*-value less than 0.01 were deemed highly significant.

## 5. Conclusions

At present, the treatments available for osteoarthritis are not sufficiently effective, and many patients with early-stage osteoarthritis show severe symptoms. Knee replacement surgery is an expensive treatment, and many patients are unable to afford it. As a result, there is a need for more effective treatment options for this condition. This study demonstrates that early intervention with intra-articular injection of MSCs is effective in preventing the progression of early-stage osteoarthritis to severe osteoarthritis. These results are superior to those of HA injections, which are commonly used to treat early-stage osteoarthritis. In addition, the injection of early-stage chondrogenic differentiated cells from MSCs, rather than MSCs alone or HA, is found to be more effective in treating knee osteoarthritis. A novel method of injecting early-stage cartilage differentiated cells from MSCs was developed in this study, which offers a superior treatment option for osteoarthritis compared to undifferentiated MSCs or HA alone. Thus, this approach could serve as a guide for the treatment of early osteoarthritis patients, where injection of chondrogenic differentiated cells into the knee joint could lead to recovery of the cartilage and prevent disease progression to severe osteoarthritis. This study demonstrates that injection of 1 × 10^6^ chondrogenic differentiated cells into the knee joints of guinea pigs was effective for early osteoarthritis treatment. However, further research is needed to optimize the treatment method for improved efficacy, including cell numbers, solvents, and number of injections. Additionally, future studies could investigate the treatment effects on moderate and severe osteoarthritis, ultimately leading to improved treatment options for human osteoarthritis patients.

## Figures and Tables

**Figure 1 ijms-25-05673-f001:**
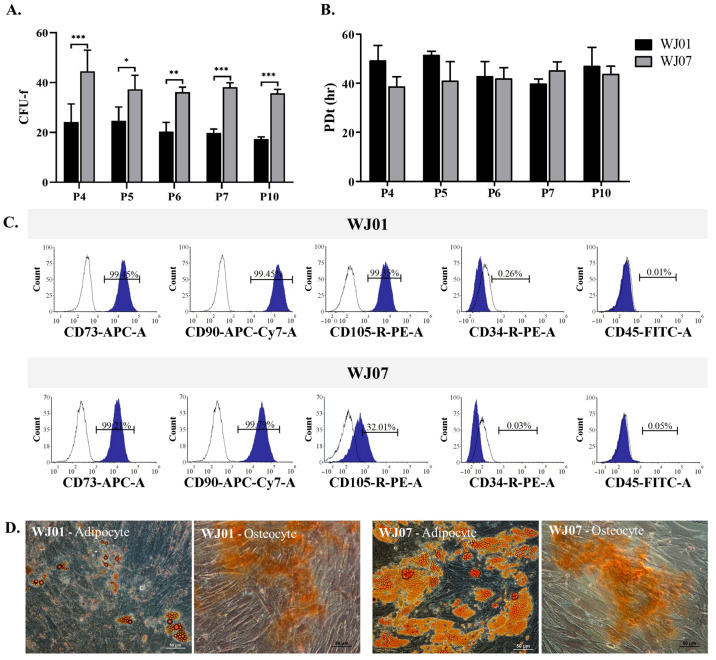
(**A**) hWJ−MSCs characterization. Colony-forming unit results from WJ01 and WJ07 cell lines at passage 4, 5, 6, 7, and 10 (* *p* < 0.05, ** *p* < 0.01, *** *p* < 0.001), (**B**) Population doubling time results of WJ01 and WJ07 cell line, (**C**) Cell surface marker expression of WJ01 and WJ07 cell lines, (**D**) Osteogenic and adipogenic differentiation of WJ01 and WJ07 cell lines. Scale bar = 50 µm.

**Figure 2 ijms-25-05673-f002:**
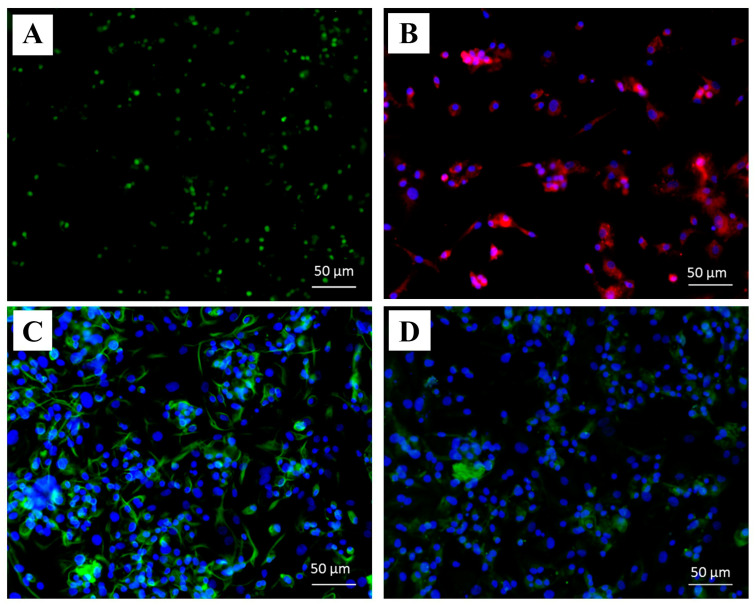
Chondrocyte characterization by ICC, (**A**) Sox9 (green; nuclear marker), (**B**) Aggrecan (red), (**C**) type II collagen (green), (**D**) type X collagen (green). Scale bar = 50 µm.

**Figure 3 ijms-25-05673-f003:**
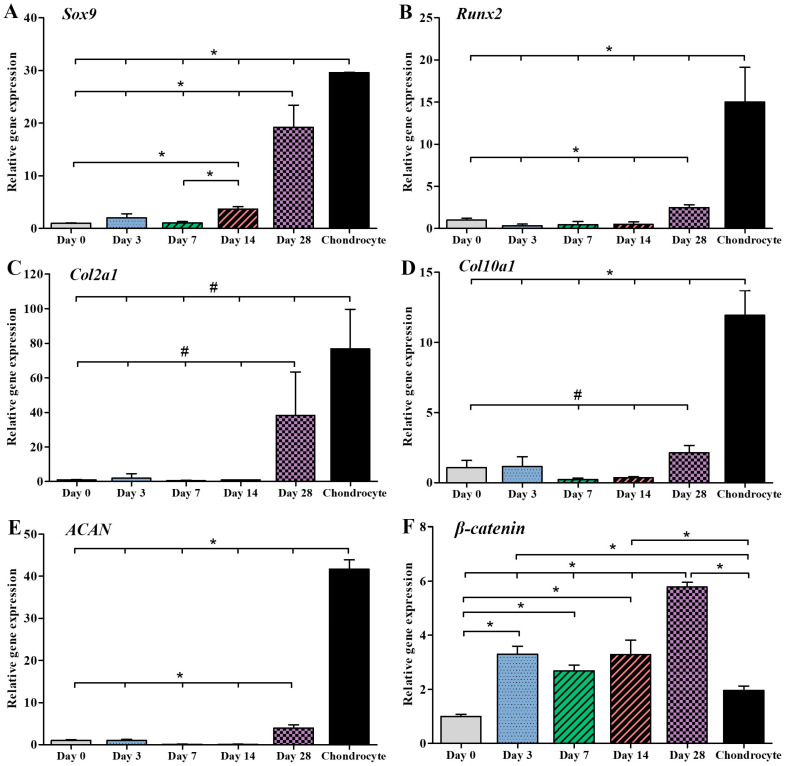
Gene expression analysis of chondrocyte by qPCR: (**A**) *Sox9*, (**B**) *Runx2*, (**C**) *Col2a1*, (**D**) *Col10a1*, (**E**) *ACAN*, and (**F**) *β-Catenin* genes. The targeted gene was normalized to *GAPDH* as a reference gene, and the relative expression was calculated and compared to each group. (* *p* < 0.01, # *p* < 0.05).

**Figure 4 ijms-25-05673-f004:**
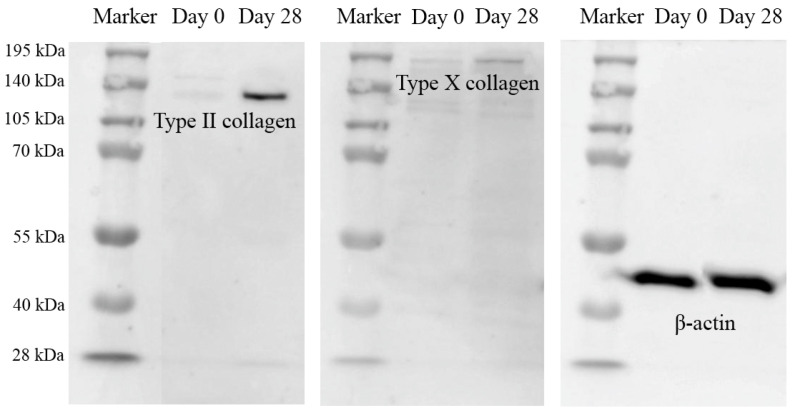
Type X collagen protein expression analysis of chondrocyte differentiated cells at day 28 by immunoblot, and β-actin protein was used as an internal control. Western blotting was carried out in duplicate.

**Figure 5 ijms-25-05673-f005:**
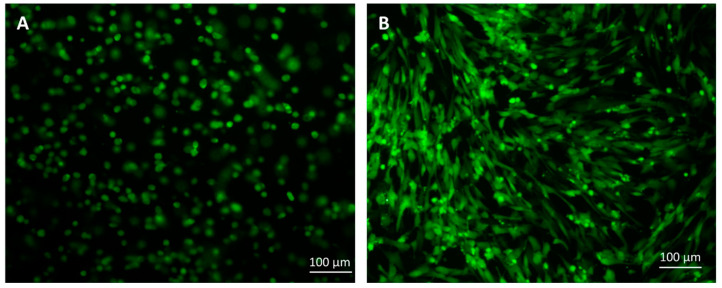
The cells stained with CFDA-SE fluorescent dye: (**A**) cells suspended in HA, (**B**) cells cultured for 7 days. Scale bar = 100 µM.

**Figure 6 ijms-25-05673-f006:**
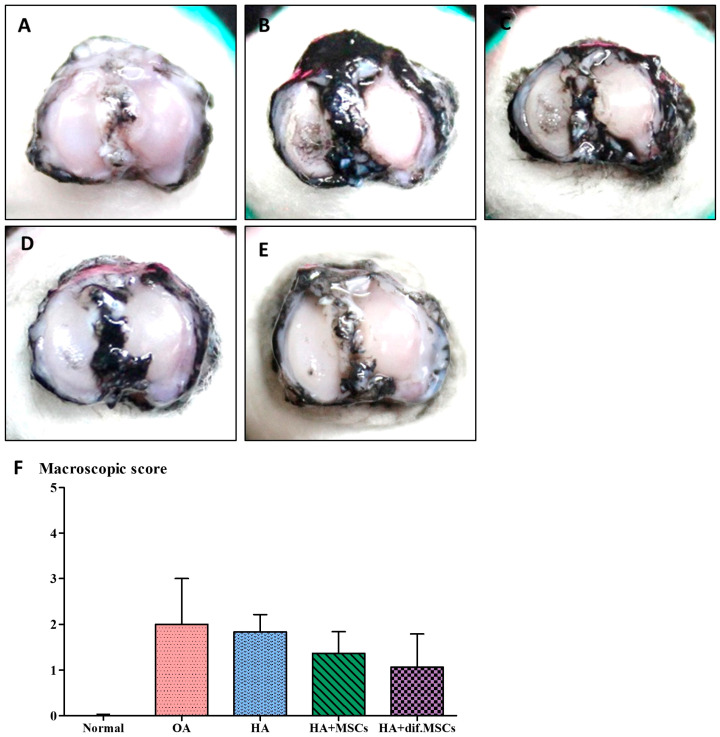
The osteoarthritis scores of each group were examined by (**A**–**E**) India ink staining and (**F**) macroscopic score.

**Figure 7 ijms-25-05673-f007:**
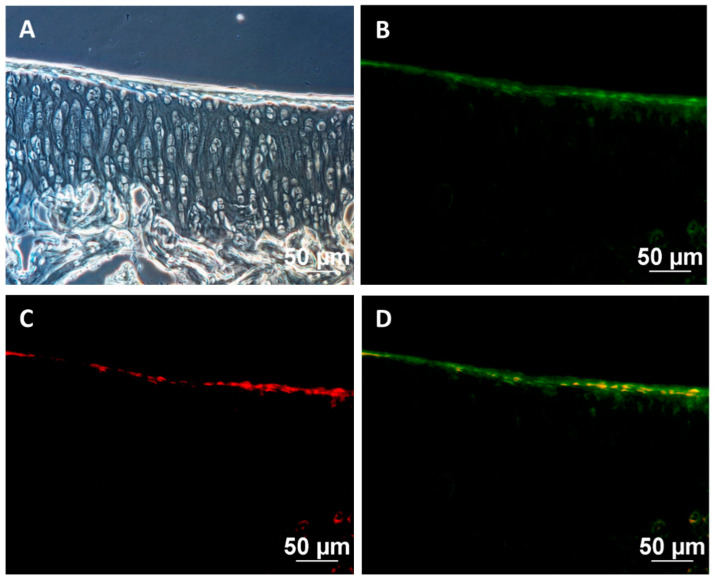
Cell tracking after transplantation: (**A**) bright field images, (**B**) CFDA-SE-stained transplanted cells (green), (**C**) human nuclei (red), (**D**) merged images. Scale bar = 50 μm.

**Figure 8 ijms-25-05673-f008:**
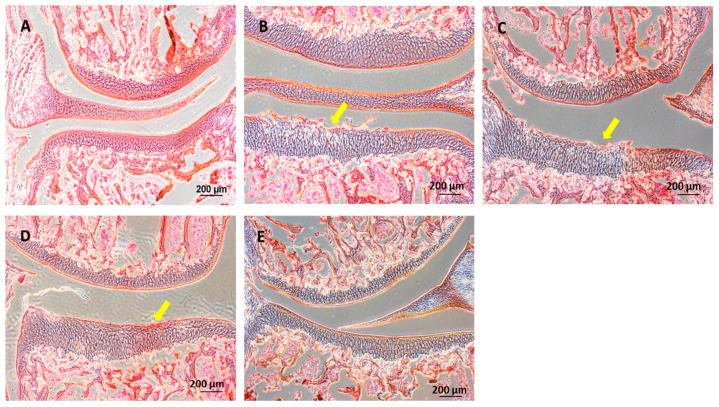
Histological examination by H&E staining: (**A**) group 1; 3 months old, (**B**) group 2; 7 months old, (**C**) group 3; 7 months old with HA injection, (**D**) group 4; 7 months old with HA + MSCs injection, (**E**) group 5; 7 months old with HA + chondrocyte differentiated cells injection. Scale bar = 200 μm.

**Figure 9 ijms-25-05673-f009:**
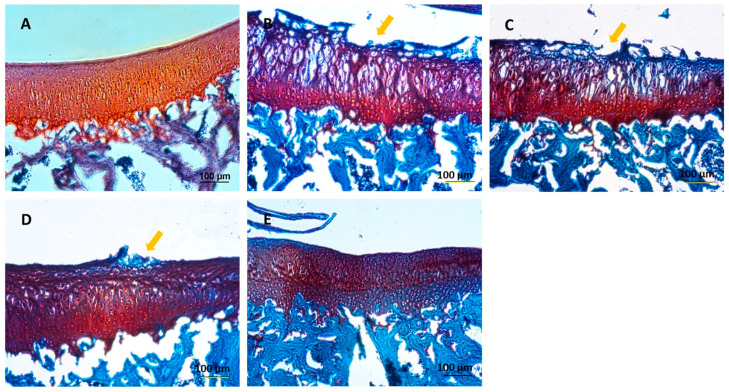
Histological examination by Safranin O staining: (**A**) group 1; 3 months old, (**B**) group 2; 7 months old, (**C**) group 3; 7 months old with HA injection, (**D**) group 4; 7 months old with HA + MSCs injection, (**E**) group 5; 7 months old with HA + chondrocyte differentiated cells injection. Scale bar = 100 μm.

**Figure 10 ijms-25-05673-f010:**
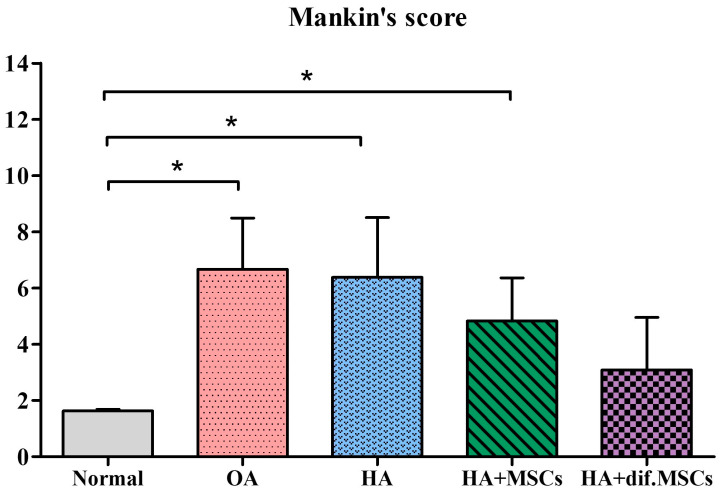
Cartilage damage scores based on the Mankin criteria (* *p* < 0.01).

**Figure 11 ijms-25-05673-f011:**
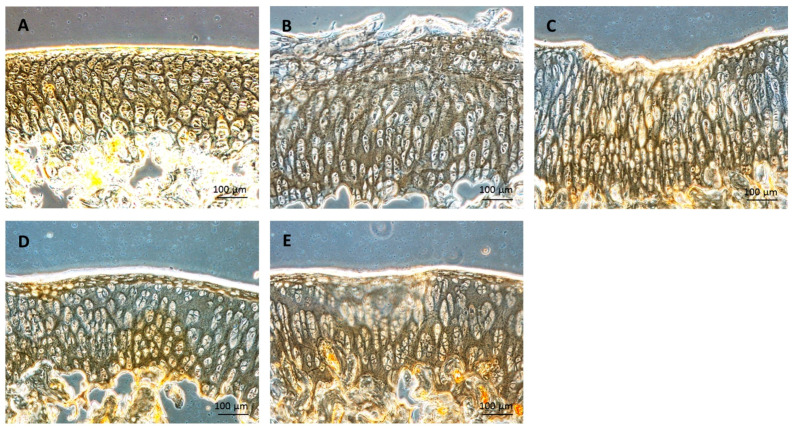
Immunohistochemistry for type II collagen: (**A**) group 1; 3 months old, (**B**) group 2; 7 months old, (**C**) group 3; 7 months old with HA injection, (**D**) group 4; 7 months old with HA + MSCs injection, (**E**) group 5; 7 months old with HA + chondrocyte differentiated cells injection. Scale bar = 100 μm.

**Figure 12 ijms-25-05673-f012:**
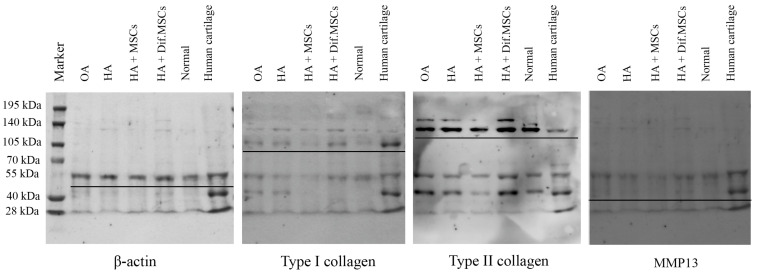
Immunoblot analysis after protein bands were isolated by gel electrophoresis.

**Figure 13 ijms-25-05673-f013:**
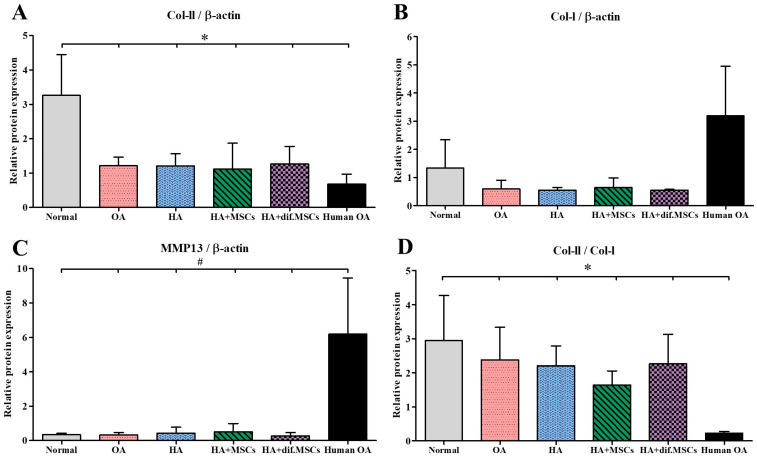
(**A**) Intensity changes in type II collagen, (**B**) type I collagen, and (**C**) MMP13 proteins in guinea pig cartilage and human cartilage with osteoarthritis, compared with β-actin protein used as an internal control (* *p* < 0.01, # *p* < 0.05), and (**D**) intensity changes in type II collagen/type I collagen in guinea pig cartilage and human cartilage with osteoarthritis.Intensity changes in Col-II/β-actin in the cartilage of the normal group were significantly higher than other groups (*p* < 0.01), but there was no difference among OA, HA, HA+MSCs, HA+dif.MSCs, and Human OA groups. The results of the intensity changes in Col-I/β-actin, on the other hand, revealed no difference in cartilage in all groups. For the results of human cartilage with severe osteoarthritis (Human OA), the intensity changes in Col-I/β-actin in Human OA were greater than in all other groups. MMP13/β-actin intensity changes were similarly low in all guinea pig groups and significantly (*p* < 0.05) lower than in the Human OA group. The highest Col-II/Col-I protein expression was seen in the normal group. Col-II/Col-I protein expression in the Human OA group was very low and significantly (*p* < 0.01) lower than those in guinea pigs in all experimental groups.

**Figure 14 ijms-25-05673-f014:**
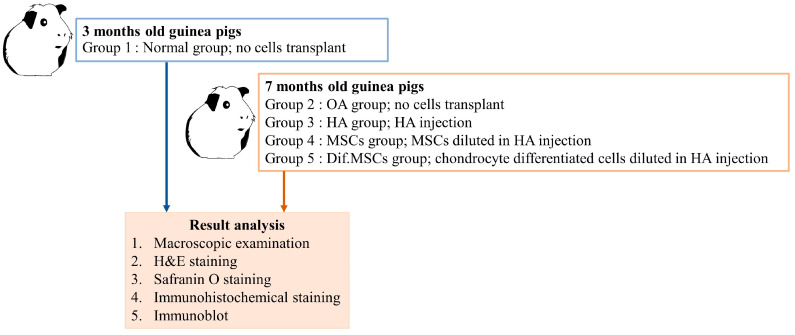
Experimental design of cell transplantation.

**Table 1 ijms-25-05673-t001:** Primers used for gene expression analysis.

Genes	Accession Number	Primer Sequence (5′–3′)	Product Size (bp)	References
*ACAN*	001113455.1	F: ACTTCCGCTGGTCAGATGGA R: TCTCGTGCCAGATCATCACC	111	[36]
*Sox9*	000346.4	F: ACACACAGCTCACTCGACCTTG R: GGGAATTCTGGTTGGTCCTCT	103	[37]
*Col2a1*	001844.4	F: GAGACAGCATGACGCCGAG R: GCGGATGCTCTCAATCTGGT	67	[38]
*Col10a1*	000493.4	F: CCCTCTTGTTAGTGCCAACC R: AGATTCCAGTCCTTGGGTCA	155	[23]
*Runx2*	001015051.4	F: ATACCGAGTGACTTTAGGGATGC R: AGTGAGGGTGGAGGGAAGAAG	131	[23]
*β-catenin*	001330729.2	F: AATGCTTGGTTCACCAGTG R: GGCAGTCTGTCGTAATAGCC	176	[23]
*GAPDH **	002046.7	F: TGCACCACCACCTGCTTAGC R: GGCATGGACTGTGGTCATGAG	87	[38]

* Reference gene.

**Table 2 ijms-25-05673-t002:** Scoring criteria for osteoarthritis symptoms according to cartilage damage examined by India ink staining.

Score	Cartilage Surface
0	normal, perfectly smooth surface, no black areas
1	small area of rough surface, only stick black on small area <10%
2	medium area of rough surface, stick black on small area 10–30%
3	large area of rough surface, stick black on wide and dark area >30%
4	Cartilage loss areas are deep but not damaged to the bone
5	Cartilage loss areas are deep and damaged to the bone

## Data Availability

All data generated or analyzed during this study are included in this published article. Further requests are welcomed, and please contact the corresponding authors.

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
