# Peer review of "Induction of Human Wharton’s Jelly of Umbilical Cord Derived Mesenchymal Stem Cells to Be Chondrocytes and Transplantation in Guinea Pig Model with Spontaneous Osteoarthritis"

_ijms, 2024, doi:10.3390/ijms25115673_

Round 1
Reviewer 1 Report
Comments and Suggestions for Authors
Generally, a nice paper supported with an extensive amount of methodology techniques and results.
I am just wondering if was not really possible include more than one donor WJ cell line -e.g. at least from 3 donors, to support more your results. If not, could be also interesting use another source of MSC - e.g. adipose tissue, or cord blood. Did you also try another sample - mentioned as WJ07 and data was not shown, or that sample was totally excluded because of low CD105 expression? Did you try also different Ab clones for CD105 detection, or different titers? What was the viability of WJ-MSC cells after thawing? Was the same for both samples 01 and 07? I am asking in order exclude other reasons of low CD105 expression of WJ07, which otherwise showed nice CFU-f and also differentional potential...
My other comment is to the statistical presentation of results in general. I think they should be re-checked. I am missing in all graphs also minimal values, besides maximal and mean - which makes difficult to analyze for reviewer. Also, the P values are mentioned only for some group while for others not (e.g. Figure 13 or Figure 3). Futhermore, e.g. in text line 128-130 you mentioned "However, level of β-catenin gene was significantly higher in cells at day 28 than them at day 14 and the positive control (chondrocyte) but not different between group day 28 with chondrocyte group. (P<0.01) - but at least to the picture seems that there is a difference also. The same also for figure 10 - I would say that there could be statistically significant difference also between normal and HA+dif.MSCs population (P<0.05) - at least if you would compare in the same way also HA+dif.MSCs and HA+MSCs. Again would be more relevant if was tested on more "cell lines".
Regarding discussion, I would try to discuss the results more and try to search for more current research - in general I see in references only 1 from 2020 and 2 from 2019, all other were older than 5 years. Maybe you could add also comparison to research used other animal models (rabbit, horse, pig..). For less oriented reader I would mention sentence in line 289 also in introduction - to better understand difference between 3 and 7 month-old guinea-pigs in the meaning of developing of osteoarthritis.
Comments on the Quality of English Language
just minor editing required
Author Response
May 9, 2024
Dear Prof. Dr. Maurizio Battino,
Editor-in-Chief of International Journal of Molecular Sciences
This is the answer to reviewers of manuscript ID: ijms-2994781 entitled “Induction of human Wharton’s jelly of umbilical cord derived mesenchymal stem cells to be chondrocytes and transplantation in guinea pig model with spontaneous osteoarthritis” submitted to International Journal of Molecular Sciences. Detailed response to the comments has been explained upon a point-by-point basis. All revisions made regarding addition and edition of information in the revised manuscript are shown with the dark blue highlighted text. The revision in response to the comments suggested by the reviewers is present as follows:
Reviewer #1
Generally, a nice paper supported with an extensive amount of methodology techniques and results.
1. I am just wondering if was not really possible include more than one donor WJ cell line -e.g. at least from 3 donors, to support more your results. If not, could be also interesting use another source of MSC - e.g. adipose tissue, or cord blood. Did you also try another sample - mentioned as WJ07 and data was not shown, or that sample was totally excluded because of low CD105 expression? Did you try also different Ab clones for CD105 detection, or different titers? What was the viability of WJ-MSC cells after thawing? Was the same for both samples 01 and 07? I am asking in order exclude other reasons of low CD105 expression of WJ07, which otherwise showed nice CFU-f and also differentional potential...
Answer: This study was performed during the covid-19 pandemic. Therefore, we had limited timeframe to collect our samples. We could get only two umbilical cords from the hospital.
Our research project focused mainly on the use of MSCs derived from umbilical cord. The other sources of MSCs as you mentioned are interested. We would like to do new project with your point of views in the future.
WJ01 and WJ07 had similar viability after thawing. We believe that CD105 works well in the present study because we simultaneously used CD105 in another project. High expression of CD105 was seen in our study. We thought that WJ107 had naturally low expression of CD105.
2. My other comment is to the statistical presentation of results in general. I think they should be re-checked. I am missing in all graphs also minimal values, besides maximal and mean - which makes difficult to analyze for reviewer. Also, the P values are mentioned only for some group while for others not (e.g. Figure 13 or Figure 3). Furthermore, e.g. in text line 128-130 you mentioned "However, level of β-catenin gene was significantly higher in cells at day 28 than them at day 14 and the positive control (chondrocyte) but not different between group day 28 with chondrocyte group. (P<0.01) - but at least to the picture seems that there is a difference also. The same also for figure 10 - I would say that there could be statistically significant difference also between normal and HA+dif.MSCs population (P<0.05) - at least if you would compare in the same way also HA+dif.MSCs and HA+MSCs. Again would be more relevant if was tested on more "cell lines".
Answer: In figure 3 and 13, we have re-checked our results and re-tested statistical analyze at P-value of 0.01 and 0.05. Based on the results of re- analysis, we found significant differences of the expression of gene markers monitored in the present study. There were a few differences of the results present in the original manuscript. We found that β-catenin gene expression of chondrogenic differentiated cells at day 3, 7, and 14 was not significantly (P>0.01) different, but was significantly (P<0.01) higher than undifferentiated MSCs (day 0). Level of β-catenin gene significantly (P<0.01) increased in the cells at day 28, compared to the cells at day 0, 3, 7, 14 and the positive control (chondrocyte). Moreover, β-catenin gene expression of the cells at day 3 and 14 was also significantly (P<0.01) higher than the positive control. Expression of Sox9 gene was significantly higher in induced chondrogenic cells at day 28 than at day 0-14 (P< 0.01). At day 3, 7 and 14 of induction, Runx2, Col2a1, Col10a1 and ACAN gene expression showed no significant difference among groups. Until day 28, the expression of Runx2, Col2a1, Col10a1, and ACAN genes was significantly higher than the undifferentiated MSCs (day 0), but still significantly lower than the positive control.
In Figure 13, we found that intensity changes of Col-II/β-actin in the cartilage of normal group were significantly higher than other groups (P< 0.01), but there was no difference among OA, HA, HA+MSCs, HA+dif.MSCs and Human OA groups. The results of the intensity changes of Col-I/β-actin, on the other hand, revealed no difference in cartilage in all groups. For the results of human cartilage with severe osteoarthritis (Human OA), the intensity changes of Col-I/β-actin in Human OA were greater than in all other groups. MMP13/β-actin intensity changes were similarly low in all guinea pig groups and significantly (P<0.05) lower than in Human OA group. The highest Col-II/Col-I protein expression was seen in normal group. Col-II/Col-I protein expression in Human OA group was very low and significantly (P<0.01) lower than those in guinea pigs in all experimental groups. Changes can be found in new figure 3 and 13, and at line 139-149, 188-190 and 264-277.
For figure 10, we have re-checked statistical analyses of the results in the present study. The results were similar to those present in the original manuscript as follows. Significant differences of cartilage damage scores were found between normal and OA, normal and HA, and normal and HA+MSCs at P-value of 0.01. However, there were no significant differences of cartilage damage scores observed in guinea pigs treated with HA+MSCs and HA+ dif.MSCs at P-value of 0.01. Consequently, we did not add symbol (*) in the figure 10. We marked the symbol (*) when only significant difference was found between treatments. In this figure, statistical analyze were conducted at P-value of 0.01 in order to receive highly accurate results.
3. Regarding discussion, I would try to discuss the results more and try to search for more current research - in general I see in references only 1 from 2020 and 2 from 2019, all other were older than 5 years. Maybe you could add also comparison to research used other animal models (rabbit, horse, pig..). For less oriented reader I would mention sentence in line 289 also in introduction - to better understand difference between 3 and 7 month-old guinea-pigs in the meaning of developing of osteoarthritis.
Answer: Thank you very much for your suggested and provided information for discussion. We have revised Discussion section and incorporated updated references in the manuscript. Change can be found at line 332-338.

Reviewer 2 Report
Comments and Suggestions for Authors The manuscript presents a novel and potentially transformative approach to the treatment of osteoarthritis by harnessing the regenerative capabilities of Wharton's jelly-derived MSCs.
Some essential comments:
1. Lines 23-24: 'Results showed that hWJ-MSCs-derived chondrocytes expressed specific markers of chondrocytes.' - Sounds primitive, I recommend listing the markers. 2. Lines 37-38: 'The disease can be triggered by genetic and non-genetic factors related to the...' - this sentence should be extended. 3. Lines 35-86: The Introduction section consists of simply and extremely short sentences and should be fully-rewritten. What was the aim of this study? 4. Line 89: 'hWJ-MSCs obtained from two umbilical cords (WJ01 and WJ07).' - the authors should disclose the source of these cells and biobanking repository identifiers. 5. Lines 254-303: Results and Discussion: While the study demonstrates the effectiveness of the treatment, it does not delve deeply into the mechanisms by which hWJ-MSCs-derived chondrocytes promote cartilage recovery. For example, transplantation of cultured chondrocytes can induce variable inflammatory responses involving mast cells and macrophages [https://pubmed.ncbi.nlm.nih.gov/38069106/]. Why did the authors not describe or even mention related to inflammation reactions in their study? Did the authors suppose that inflammatory responses had no place in their study? 6. Line 426: The paper does not provide details on the sample size of guinea pigs used in each treatment group. 7. Line 426: "4.4.1. Experimental animals" - There is a need to detail the type of animal marking. It should be noted that some types of animal identifications could affect the outcomes. 8. Line 491: Why do the authors use ANOVAs to compare two groups instead of three or more? 9. Why didn't the authors do an ultrasound or CT scan to confirm functional outcomes? Figures: 10. Figures 1-5 are superfluous, they describe the first steps of the study and will be merged into 2 combined figures. 11. Figures 8,9,11: Why do the authors use so few histological techniques (H&E, Safranin O, type II collagen IHC? 12. Figure 7: What conclusions can be drawn from this figure, other than the known fact that injected chondrocytes spread over cartilage? Tables 13. Table 1: Scoring criteria - it should be in the Methods.
Author Response
May 9, 2024
Dear Prof. Dr. Maurizio Battino,
Editor-in-Chief of International Journal of Molecular Sciences
This is the answer to reviewers of manuscript ID: ijms-2994781 entitled “Induction of human Wharton’s jelly of umbilical cord derived mesenchymal stem cells to be chondrocytes and transplantation in guinea pig model with spontaneous osteoarthritis” submitted to International Journal of Molecular Sciences. Detailed response to the comments has been explained upon a point-by-point basis. All revisions made regarding addition and edition of information in the revised manuscript are shown with the dark blue highlighted text. The revision in response to the comments suggested by the reviewers is present as follows:
Reviewer #2
The manuscript presents a novel and potentially transformative approach to the treatment of osteoarthritis by harnessing the regenerative capabilities of Wharton's jelly-derived MSCs.
Some essential comments:
1. Lines 23-24: 'Results showed that hWJ-MSCs-derived chondrocytes expressed specific markers of chondrocytes.' - Sounds primitive, I recommend listing the markers.
Answer: We have revised by incorporating the specific gene markers that expressed in hWJ-MSCs-derived chondrocytes following your suggestion. Change can be found at line 25-26.
2. Lines 37-38: 'The disease can be triggered by genetic and non-genetic factors related to the...' - this sentence should be extended.
Answer: We have improved the sentence following your suggestion. Change can be found at line 44-47.
3. Lines 35-86: The Introduction section consists of simply and extremely short sentences and should be fully-rewritten. What was the aim of this study?
Answer: We have improved the Introduction section and addressed the aims of the present study in the revised manuscript. Changes can be found at line 59-70 and 101-105.
4. Line 89: 'hWJ-MSCs obtained from two umbilical cords (WJ01 and WJ07).' - the authors should disclose the source of these cells and biobanking repository identifiers.
Answer: The hWJ-MSCs used in the present study were obtained from umbilical cords freshly collected under a written informed consent of the patients at Maharat Nakhon Ratchasima Hospital, Thailand. Please see subsection 2.1: isolation and characterization of MSCs. Change can be found at line 108-109 and subsection 4.2: hWJ-MSCs isolation and culture in Materials and Methods, line 356-357.
5. Lines 254-303: Results and Discussion: While the study demonstrates the effectiveness of the treatment, it does not delve deeply into the mechanisms by which hWJ-MSCs-derived chondrocytes promote cartilage recovery. For example, transplantation of cultured chondrocytes can induce variable inflammatory responses involving mast cells and macrophages [https://pubmed.ncbi.nlm.nih.gov/38069106/]. Why did the authors not describe or even mention related to inflammation reactions in their study? Did the authors suppose that inflammatory responses had no place in their study?
Answer: Thank you for your suggestion. In the present study, we did not study about inflammatory reactions and deep mechanisms of hWJ-MSCs-derived chondrocytes. The present experiment is preliminary research. The results of this study showed that OA cartilages can be regenerated after MSCs injection. Therefore, in the future, we will study deeply into the mechanisms by MSCs in cartilage regeneration. We have discussed the issue related to inflammatory responses after MSCs transplantation in line 338 - 344 of the revised manuscript following your suggestion.
6. Line 426: The paper does not provide details on the sample size of guinea pigs used in each treatment group.
Answer: In the present study, we used 10 guinea pigs per treatments as present in subsection 4.4.1 experimental animals, line 475.
7. Line 426: "4.4.1. Experimental animals" - There is a need to detail the type of animal marking. It should be noted that some types of animal identifications could affect the outcomes.
Answer: We honestly appreciate your suggestion. We marked the guinea pigs using ear notching technique (please see subsection 4.4.1 experimental animals. Change can be found at line 482-483.
8. Line 491: Why do the authors use ANOVAs to compare two groups instead of three or more?
Answer: We honestly apologize for the misleading sentence. ANOVA was employed to find the differences between the control and treated (group 2-5) groups. Change can be found at line 545-546.
9. Why didn't the authors do an ultrasound or CT scan to confirm functional outcomes?
Answer: We did not conduct an ultrasound and CT scan because of our university does not have ultrasound and CT scan for animals and we also had limited budget subsidized from funding agency. We hope this would not weaken the quality of the present study and the manuscript can be published in this journal.
10. Figures 1-5 are superfluous, they describe the first steps of the study and will be merged into 2 combined figures.
Answer: Thank you for your suggestion. In the present study, we studied a broad spectrum of hWJ-MSCs-derived chondrocytes in order to thoroughly explain their characteristics prior to transplantation into OA-suffered guinea pigs. We would like to describe the characteristics of hWJ-MSCs-derived chondrocytes upon a point-by-point basis. Consequently, we thought the figures present in the manuscript are important evidence to illustrate all detailed and apparent characteristics of hWJ-MSCs-derived chondrocytes. This is also useful to provide a complete picture of the present investigation. Moreover, there is no limited page for publication in the journal.
11. Figures 8,9,11: Why do the authors use so few histological techniques (H&E, Safranin O, type II collagen IHC?
Answer: The three standard histological and histochemical techniques in this study are widely used to examine the recovery progress of cartilage transplanted with stem cells in animal models, and cartilage tissue damages (as listed below). In the present study, we aimed to investigate cartilage tissue damage and functional performance of the hWJ-MSCs-derived chondrocytes after implantation in guinea pigs. The results obtained from the three techniques deciphered all our purposes in terms of cartilage damage and function in the present study.
Pauli et al. (2012). Comparison of cartilage histopathology assessment systems on human knee joints at all stages of osteoarthritis development. Osteoarthritis and Cartilage, 20, 476-485.
Zhang et al. (2013). Neonatal desensitization supports long-term survival and functional integration of human embryonic stem cell-derived mesenchymal stem cell in rat joint cartilage without immunosuppression. Stem Cells and Development, 22, 90-101.
Abe et al. (2023).Engraftment of allogenetic iPC cell-derived cartilage organoid in a primate model of articular cartilage defect. Nature Communications, 14(1), 804.
12. Figure 7: What conclusions can be drawn from this figure, other than the known fact that injected chondrocytes spread over cartilage?
Answer: We have incorporated an implication obtained from the results that are illustrated in Fig 7. Change can be found at line 209-211.
13. Table 1: Scoring criteria - it should be in the Methods.
Answer: We have moved the Table 1: scoring criteria to Materials and Methods section and renamed as Table 2 following your suggestion. Change can be found at line 510-512.

Reviewer 3 Report
Comments and Suggestions for Authors
In the article: “Induction of Human Wharton’s Jelly of Umbilical Cord Derived Mesenchymal Stem Cells to Be Chondrocytes and Transplantation in Guinea Pig Model with Spontaneous Osteoarthritis”, the authors focused on differentiation of human Wharton’s jelly-derived mesenchymal stem cells (hWJ-MSCs) into chondrocytes for transplantation in OA-suffered guinea pigs.
This manuscript appears interesting, the authors clearly explain the rational of the study and discussed the topic point by point. Anyway, we would like to invite the authors to better clarify some points:
1. Please check the check punctuation and spaces;
2. Within the introduction, the authors do not clearly explain the current approaches for OA treatment. Please, try to briefly introduce the concept of pharmacological and no pharmacological therapies with relative examples (food supplements, specific drugs ..);
3. The authors explain the importance of HA for the implants of cells, anyway, they do not explain in general HA importance and several application for OA therapies. In this respect, the following reference should be useful: “La Gatta A, Stellavato A, Vassallo V, Di Meo C, Toro G, Iolascon G, Schiraldi C. Hyaluronan and Derivatives: An In Vitro Multilevel Assessment of Their Potential in Viscosupplementation. Polymers (Basel). 2021 Sep 22;13(19):3208. doi: 10.3390/polym13193208. PMID: 34641024; PMCID: PMC8512809”;
4. Concerning “hWJ-MSCs isolation and culture” protocol, is there available a specific reference?
5. Did you try to observe, a “spontaneous” differentiation of cells without specific culture medium?
6. Figures: please in the figures caption specify the used magnification;
7. Figure 4: please specify the number of performed western blotting. Moreover, densitometric analyses of these results should be useful;
8. Figure 6: in the graph “normal” related bar is not visible;
9. Figure 12: the quality of western images is very low, do you have better acquisitions?
Comments on the Quality of English Language
In this manuscript minor spelling and editing mistakes are present.
Author Response
May 9, 2024
Dear Prof. Dr. Maurizio Battino,
Editor-in-Chief of International Journal of Molecular Sciences
This is the answer to reviewers of manuscript ID: ijms-2994781 entitled “Induction of human Wharton’s jelly of umbilical cord derived mesenchymal stem cells to be chondrocytes and transplantation in guinea pig model with spontaneous osteoarthritis” submitted to International Journal of Molecular Sciences. Detailed response to the comments has been explained upon a point-by-point basis. All revisions made regarding addition and edition of information in the revised manuscript are shown with the dark blue highlighted text. The revision in response to the comments suggested by the reviewers is present as follows:
Reviewer #3
In the article: “Induction of Human Wharton’s Jelly of Umbilical Cord Derived Mesenchymal Stem Cells to Be Chondrocytes and Transplantation in Guinea Pig Model with Spontaneous Osteoarthritis”, the authors focused on differentiation of human Wharton’s jelly-derived mesenchymal stem cells (hWJ-MSCs) into chondrocytes for transplantation in OA-suffered guinea pigs.
This manuscript appears interesting, the authors clearly explain the rational of the study and discussed the topic point by point. Anyway, we would like to invite the authors to better clarify some points:
1. Please check the check punctuation and spaces
Answer: We have thoroughly read and carefully revised by checking punctuation and spaces in the revised manuscript.
2. Within the introduction, the authors do not clearly explain the current approaches for OA treatment. Please, try to briefly introduce the concept of pharmacological and no pharmacological therapies with relative examples (food supplements, specific drugs ..)
Answer: We have revised the Introduction section by introducing the concept of pharmacological and non-pharmacological treatments for OA patients. Change can be found at line 59-70.
3. The authors explain the importance of HA for the implants of cells, anyway, they do not explain in general HA importance and several application for OA therapies. In this respect, the following reference should be useful: “La Gatta A, Stellavato A, Vassallo V, Di Meo C, Toro G, Iolascon G, Schiraldi C. Hyaluronan and Derivatives: An In Vitro Multilevel Assessment of Their Potential in Viscosupplementation. Polymers (Basel). 2021 Sep 22;13(19):3208. doi: 10.3390/polym13193208. PMID: 34641024; PMCID: PMC8512809”
Answer: Thank you for your suggestion. We have revised the Introduction and Discussion sections by incorporation of HA importance and application for OA treatment. Changes can be found at line 91-93 and 319-324.
4. Concerning “hWJ-MSCs isolation and culture” protocol, is there available a specific reference?
Answer: hWJ-MSCs isolation and culture were conducted following a protocol described by Tanthaisong et al. [22]. Change can be found at line 356-357.
5. Did you try to observe, a “spontaneous” differentiation of cells without specific culture medium?
Answer: On a basis of our experience, we have never observed a spontaneous differentiation of hWJ-MSCs in our culture medium. However, in this study, during the examination of the population doubling time of hWJ-MSCs, we found no spontaneous differentiation.
6. Figures: please in the figures caption specify the used magnification
Answer: We have added scale bars in all figures and figure captions in the manuscript. We believe that the magnification is not necessary in the figure captions.
7. Figure 4: please specify the number of performed western blotting. Moreover, densitometric analyses of these results should be useful
Answer: Western blotting was carried out in duplicate. Please see Figure 4 caption, line 193-194.
8. Figure 6: in the graph “normal” related bar is not visible
Answer: We have replaced old figure with new revised figure.
9. Figure 12: the quality of western images is very low, do you have better acquisitions?
Answer: We apologize to say that this is the best figure in our hands. We have tried to improve its quality by adjusting brightness and contrast of the figure. We have replaced the old figure with the adjusted figure as present in page 11, line 262 of revised manuscript.
10. Comments on the Quality of English Language
In this manuscript minor spelling and editing mistakes are present.
Answer: We have thoroughly read and carefully revised in the manuscript.

Round 2
Reviewer 1 Report
Comments and Suggestions for Authors
Thank you for accepting most of the comments.
Author Response
May 17, 2024
Dear Prof. Dr. Maurizio Battino,
Editor-in-Chief of International Journal of Molecular Sciences
This is the answer to reviewer of manuscript ID: ijms-2994781 entitled “Induction of human Wharton’s jelly of umbilical cord derived mesenchymal stem cells to be chondrocytes and transplantation in guinea pig model with spontaneous osteoarthritis” submitted to International Journal of Molecular Sciences. Detailed response to the comments has been explained upon a point-by-point basis. All revisions made regarding addition and edition of information in the revised manuscript are shown with the dark blue highlighted text as follows: